# M$^4$I: Multi-modal Models Membership Inference

**Pingyi Hu**[*]
University of Adelaide
Australia

**Zihan Wang**[*]
University of Adelaide
Australia

**Ruoxi Sun**
CSIRO's Data61
Australia

**Hu Wang**
University of Adelaide
Australia

**Minhui Xue**
CSIRO's Data61
Australia

## Abstract

With the development of machine learning techniques, the attention of research has been moved from single-modal learning to multi-modal learning, as real-world data exist in the form of different modalities. However, multi-modal models often carry more information than single-modal models and they are usually applied in sensitive scenarios, such as medical report generation or disease identification. Compared with the existing membership inference against machine learning classifiers, we focus on the problem that the input and output of the multi-modal models are in different modalities, such as image captioning. This work studies the privacy leakage of multi-modal models through the lens of membership inference attack, a process of determining whether a data record involves in the model training process or not. To achieve this, we propose Multi-modal Models Membership Inference (M$^4$I) with two attack methods to infer the membership status, named metric-based (MB) M$^4$I and feature-based (FB) M$^4$I, respectively. More specifically, MB M$^4$I adopts similarity metrics while attacking to infer target data membership. FB M$^4$I uses a pre-trained shadow multi-modal feature extractor to achieve the purpose of data inference attack by comparing the similarities from extracted input and output features. Extensive experimental results show that both attack methods can achieve strong performances. Respectively, 72.5% and 94.83% of attack success rates on average can be obtained under unrestricted scenarios. Moreover, we evaluate multiple defense mechanisms against our attacks. The source code of M$^4$I attacks is publicly available at https://github.com/MultimodalMI/Multimodal-membership-inference.git.

## 1 Introduction

Machine learning has witnessed great progress during recent years and now has been applied to many multi-modal applications (models that can process and relate information from multiple modalities, such as image, audio, and text), such as audio-visual speech recognition [1], speech recognition [1], event detection [2], and image captioning [3–13]. However, prior research [14] has revealed that machine learning models are vulnerable to membership inference attack. Leaking membership status (*i.e.*, member or non-member of a dataset) in a multi-modal application leads to serious security and privacy issues. For example, if a person, unfortunately, gets a tumor and his medical records were used to train an image captioning model (*e.g.*, to predict whether the tumor is benign or malignant), the attacker can infer that the person has a tumor. Furthermore, in a large-scale neural language model, the attacker can infer detailed private data with the membership inference attack, such as

---

[*] Contributed equally.

36th Conference on Neural Information Processing Systems (NeurIPS 2022).

Table 1: An overview of membership inference research.

| Target tasks | Attacks | Shadow Model | Detailed Model Prediction (*e.g.*, confidence vector) | Final Model Prediction (*e.g.*, label) | Other Useful Information |
|---|---|---|---|---|---|
| Classification | [14, 20, 31–38] | ● or - | ● | ● | Model weights |
| Generation models | [39–45] | ● or - | - | ● | - |
| NLP | [15, 30] | ● | ● or - | ● | Reference output |
| Embedding models | [16, 46–48] | ● | ● or - | ● or - | Inner layer output/feature |
| Federated models | [25, 49–53] | ● or - | ● or - | ● or - | - |
| Recommendation system | [54] | ● | - | ● | - |
| Multi-modal models (ours) | Metric-based attack | ● | - | ● | Reference output |
| | Feature-based attack | ● | - | ● | - |

"●": the adversary needs the knowledge; "-": the knowledge is unnecessary.

address or phone number, from the training data by text sequence generation [15]. This work studies the privacy leakage of multi-modal models through the lens of membership inference attack.

## 1.1 Background

**Membership inference.** Previous studies [14, 16–29] on membership inference mainly target the classification models trained from scratch, and many of them focus on exploiting the confidence scores returned by the target modal as shown in Table 1. Since the concept of membership inference in machine learning was established in Shokri *et al.*'s work [14], later approaches [18, 20, 30] further improved these methods by exploring different assumptions (multiple shadow models, knowledge of the training data distribution and model structure [16, 31]) about the attacker and by broadening the scenarios of membership inference attacks [14, 20]. In addition to classifier-based approaches [14, 20, 31–38], membership inference has been shown effective under multiple machine learning settings, such as generative models [39–45], natural language models [15, 30], embedding models [16, 46–48], and federated learning [25, 49–53]. However, to the best of our knowledge, to date there is no membership inference attack targeting at multi-modal models.

**Multi-modal machine learning.** Unlike single-modal machine learning, multi-modal machine learning aims to build models that can manage and relate information from two or more modalities [55]. A new category of multi-modal applications has been emerging, such as image captioning [3–13] and media description and generation [56–58].

In this research, we conduct membership inference on a multi-modal task, image captioning [59] that is the process of generating a textual description of an image [3–13]. In image captioning, the datasets consist of image-text pairs.

## 1.2 Our Work

Unlike the single-modal learning task, training a multi-modal model can make use of more sensitive data from different modalities. For instance, in the medical domain, medical report generation [60, 61, 59, 62–64] can reduce the workload of medical professionals and speed up the diagnosis process by automatically analyzing medical examinations, such as computed tomography and X-ray photographs. However, the existing membership inference models can hardly be applied to multi-modal models for the following two reasons: (*i*) the multi-modal models, cannot provide much information such as confidence scores of the model prediction. (*ii*) due to the requirements of handling multi-modal data, many existing membership inference methods, which target single-modal learning models such as image or text classification, are difficult to be squarely applied to multi-modal models. This motivates us to investigate a new category of membership inference attack on multi-modal models, which is, given access to a multi-modal data and model, an attacker can tell if a specific data sample is adopted in the model training process. In this work, we propose **m**ulti-**m**odal **m**odels **m**embership **i**nference (M$^4$I). Our main contributions are as follows:

- To the best of our knowledge, we are the first to investigate membership inference attacks on multi-modal models. To achieve this, we propose a metric-based attack and a feature-based attack to conduct M$^4$I under different assumptions.

- We conduct extensive experiments for general image captioning on MSCOCO, FLICKR 8k, and IAPR TC-12 datasets and we receive promising results. Furthermore, we apply our M$^4$I to a medical report generation model to witness a similar performance trend in the real world.

- We perform a detailed analysis of possible defense strategies, which are differential privacy, $l_2$ regularization, and data augmentation. These approaches provide avenues for future design of privacy mitigation strategies against M$^4$I.

## 2 Problem Formulation

In this section, we introduce the membership inference attack and formulate the threat model.

### 2.1 Membership Inference

Membership inference is a binary classification task. It aims to infer whether a data sample belongs to the training set of the target machine learning model. By querying the target model with a data sample, we can obtain the corresponding output. Then, the attack model takes it as the input and gives a binary output, indicating if this sample has been seen in the training set.

### 2.2 Threat Model

**Attacker's goal.** The adversary aims to infer whether an input data sample comes from the training dataset of a target image captioning model or not. If this input is in the target model's training dataset, we call it a member; otherwise, we call it a non-member. The adversary's goal is to accurately infer the member status (member/non-member of the target model's training dataset.)

**Attacker's background knowledge.** In our scenario, the attacker can query the target image captioning model, which provides lines of words as output without confidence scores. We consider this as the most general and difficult scenario for the attacker. In this research, we take training data distribution, model architecture, and ground truth reference into consideration as the attacker's other background knowledge. In particular, if an adversary knows the distribution, we assume the adversary has access to a shadow dataset with the same distribution as the target model's training set. With knowledge of model architecture, the adversary can use the same architecture for shadow model training. When the adversary does not know the target model's architecture, the adversary can assume another architecture with the same functionality for shadow model training. If the attacker knows the ground truth reference, we assume the attacker can use the text from the image-text pair dataset as ground truth. This knowledge will be utilized in metric-based membership inference. Depending on the difficulty to access the information for an attacker, we explore three different scenarios under different assumptions

- The **unrestricted** scenario, where an attacker has knowledge of both training data distribution and model architecture. This scenario is considered as a gray-box attack.
- The **data-only** scenario, when only the training data distribution is accessible for an attacker. This scenario is considered as a gray-box attack.
- The **constrained** scenario that neither knowledge of training data distribution nor model architecture is available to an attacker. This scenario is considered as a black-box attack.

## 3 Methodology

In this section, we propose the methodology of metric-based and feature-based M$^4$I attacks.

### 3.1 Metric-based M$^4$I

The key intuition for the metric-based attack is that the output of a multi-modal model with an input sampled from its training dataset should be highly similar to or same as the ground truth reference of this sample. In the cases for image-text models such as image captioning or medical report generation, many similarity metrics such as Recall-Oriented Understudy for Gisting Evaluation (ROUGE) score [65] or Bilingual evaluation understudy (BLEU) score [66] are used to evaluate the similarity of the ground truth description and the output text. Here, we focus on the distinguishability of the membership information by these metrics, *i.e.*, the member and non-member data samples achieving differently on scores. Then we can leverage the shadow model to launch a metric-based membership inference attack. The metric-based membership inference (MB M$^4$I) can be divided into

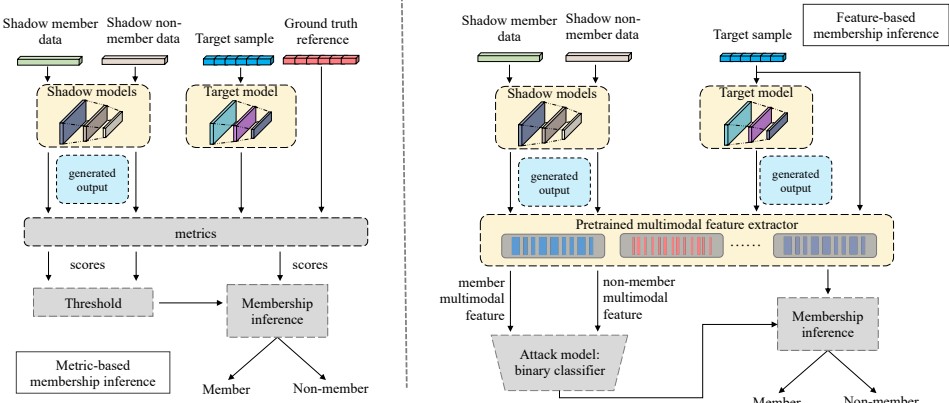

Figure 1: An overview of MB M$^4$I (left) and FB M$^4$I (right). The MB M$^4$I infers membership through similarity metric scores, while the FB M$^4$I utilizes a multi-modal feature extractor to distinguish member and non-member data on the feature level.

three main stages, namely metric calculation, preset threshold selection, and membership inference as shown in Figure 1.

**Metrics calculation.** Many metrics can be used to evaluate the similarity of the model output and corresponding ground truth. Here we utilize the ROUGE score [65] and BLEU score [66] for calculating the text-similarity for the image captioning model (see more details in Supplementary Materials).

**Preset threshold selection.** As the attacker cannot get the membership information of the training data, they can preset the threshold by training a shadow model $\mathcal{M}'$. With knowledge of the targeted task, the attacker can use a shadow dataset built up with collected public data to train a shadow model with a selected architecture. The attacker can split their shadow dataset into a member dataset and a non-member dataset and train the shadow model $\mathcal{M}'$ with the member dataset. By scoring the data from the member dataset and the non-member dataset, the adversary can easily get a threshold if there is only one metric applied. With more than one metric, the adversary can use a binary classifier to distinguish between the scores of member data and non-member data.

**Membership inference.** Based on the preset threshold, the attacker can build a threshold-based binary classifier $\mathbf{f}$ as an attack model. Then the adversary can send the target sample to the target model and get similarity scores of the output and ground truth reference by chosen metrics. The attack model can categorize the target sample as a member or a non-member.

## 3.2 Feature-based M$^4$I

Considering that the metric-based membership inference highly relies on the similarity metrics and the reference captioning of the target image, the reliability of the similarity metric can be limited in an image captioning task.

Specifically, with an image of a cat and a ground truth description, "this is a cat sitting on the road", which the model has never seen, the model might output "there is a dog on the road." However, neither the BLUE nor the ROUGE metric can give this output a high score (BLEU-1: 0.625 and ROUGE-L: 0.66) and this image can be misidentified as a member by metric-based membership inference. Additionally, as mentioned in the threat model, the adversary may not have the knowledge of the ground truth reference. To solve the drawbacks of the metric-based membership inference, we propose feature-based membership inference.

The multi-modality feature extractor is able to learn the connection of meaningful information of image and text pairs [67]. The attacker can utilize the connection of the features of the input image and the output text for the membership inference.The feature-based membership inference can be divided into three main stages: multi-modal feature extractor training, attack model training, and membership inference, as shown in Figure 1 (right).

**Multi-modal feature extractor training.** Borrowed from [67], we build a multi-modal feature extractor with two different modal encoders and a layer to align features, as shown in Figure 2. With

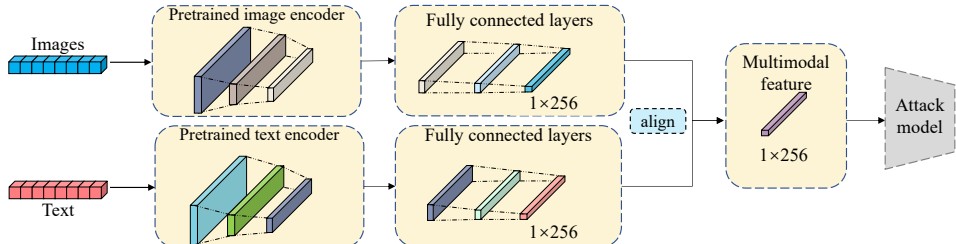

Figure 2: The multi-modal feature extractor is based on pretrained image encoder and text encoder.

some public data or a published large-scale pretrained model, the attacker can easily get a pretrained image encoder $f_i$ and a text encoder $f_t$. The encoders can be connected with a fully-connected layer so that the features of the image and text can be adjusted to the same shape. Then the attacker just needs to fine-tune the added layer by shadow dataset so that the features from the image encoder and text encoder can be aligned in the feature space.

For the $i^{th}$ sample(image $\mathbf{i}_i$ and text $\mathbf{t}_i$ pair), the feature can be formulated as follows:

$$\mathbf{F}_i = f_i(i_i); \mathbf{F}_t = f_t(t_i), \quad \mathbf{z}_i = \mathbf{F}_i - \mathbf{F}_t, \tag{1}$$

where $\mathbf{F}_i$ and $\mathbf{F}_t$ represent the image feature and text feature, respectively. $\mathbf{z}_i$ represents the difference vector between multi-modal features of the image and text.

*Parameter Optimization.* Here we use stochastic gradient descent to update parameters in the multi-modal feature extractor (MFE), aiming to minimize the Euclidean distances between image features and text features:

$$L_{MFE} = \frac{1}{N} \sum_{i=1}^{N} ||\mathbf{z}_i||_2^2, \tag{2}$$

where $|| \cdot ||_2$ denotes $L_2$ norm and $N$ is the number of training samples.

The goal of this loss function is to force the image encoder and text encoder to learn the corresponding feature representation from the image-text pairs. For each pair of images and texts, the multi-modal feature extractor would reduce the distance of their features in the same feature space.

**Attack model training.** An adversary can build an attack dataset of member multi-modal features and non-member multi-modal features using a shadow model. Those multi-modal features from the shadow member dataset can be labeled as "member" and those from the shadow non-member dataset can be labeled as "nonmember". With the attack dataset, the adversary can establish a binary classifier as the attack model $\mathcal{M}_{attack}$. The input of $\mathcal{M}_{attack}$ is $\mathbf{z}_i$ and the output of $\mathcal{M}_{attack}$ is a probability value between 0 and 1 for the membership status (0 means the attack model $\mathbf{z}_i$ is a non-member data and 1 means a member). For the $i^{th}$ sample, the prediction can be formulated as the following function:

$$\mathbf{y}_i = \mathcal{M}_{attack}(\mathbf{z}_i), \tag{3}$$

where $\mathbf{z}_i$ is the input of $\mathcal{M}_{attack}$ as well as the $i^{th}$ sample's multi-modal feature vector in our attack.

**Membership inference.** Test data $i_{test}$ for the attack model will be sent to the target model to obtain the generated captioning $t_{test}$. Then we send the $i_{test}$ and $t_{test}$ pair to the multi-modal feature extractor to get the multi-modal feature $\mathbf{z}_{test}$. The trained attack model $\mathcal{M}_{attack}$ conducts a prediction given this multi-modal feature vector $\mathbf{z}_{test}$, i.e., $\mathcal{M}_{attack}(\mathbf{z}_{test}) = \mathbf{y}_{test}$, where $\mathbf{y}_{test}$ is a value indicating the probability that the test image belongs to members. According to the predicted results, the adversary infers the membership status of the test image. Specifically, when $\mathbf{y}_{test} > 0.5$, the test image is predicted to be a member. Otherwise, it is predicted to be a non-member.

## 4  Experiments

In this section, we evaluate our metric-based and feature-based M$^4$Is on image captioning models pre-trained on different image-text datasets.

### 4.1 Experiment Setup

**Target model architectures.** The target model is an image captioning machine learning model with an encoder-decoder architecture [3]. It is used to convert a given input image $I$ into a natural language description $T$. This architecture involves using CNN layers (encoder) for feature extraction on input data, and LSTM layers (decoder) to perform sequence prediction on the feature vectors. In this work, we use Resnet-152[68] and VGG-16 [69] as heterogeneous encoders for different target model architectures.

**Datasets.** Our experiments are conducted on three different datasets: MSCOCO [70], FLICKR8k [71], and IAPR TC-12 [72], as detailed in Supplementary Materials.

**Training target model.** Our training method follows the settings in the work [3], excluding encoder architecture. We randomly sample 3,000 image-text pairs from each dataset as the "member dataset" to train the target image captioning model. Then, we randomly sample 3,000 image-text pairs from the rest of the datasets as the ground truth "non-member dataset" of the target model. Therefore, for each target image captioning model, we have 3,000 ground truth members and 3,000 ground truth non-members. We use Resnet-152 as well as VGG-16, with LSTM as the architecture for the target models.

**Training shadow models.** For both proposed M$^4$I attack methods, shadow models are indispensable. In the scenario where the attacker knows the target models' training data distribution, we randomly sample 6,000 image-text pairs from the corresponding dataset as the shadow dataset. In the scenario where the attacker does not have the knowledge of the data distribution, we randomly sample 6,000 image-text pairs from the other dataset. For instance, if the target model is trained with FLICKR8k dataset, we randomly sample 6,000 image-text pairs from the IAPR-TC12 dataset With these 6,000 image-text pairs, the attacker can split them into two 3,000 image-text datasets, which can be defined as the shadow member dataset and the shadow non-member dataset. The shadow member dataset will be used to train the shadow model and the shadow non-member dataset will be used in membership inference for attack model training.

**Notations.** To clarify the experimental settings, we use 2-letter and 4-letter combinations. For the 2-letter combinations, the first letter, *i.e.*, "C," "F" or "I", indicates the shadow (or target) dataset(which are MSCOCO, Flickr 8k, and IAPR dataset), and the second letter, *i.e.*, "R" and "V", indicates the training algorithm (which are Resnet-LSTM and VGG-LSTM). For example, "CR" is the combination of "MSCOCO+Resnet-LSTM". For the 4-letter combinations, the first two letters represent the dataset and algorithm of the shadow model and the last two letters denote the dataset and algorithm of the target model. For instance, "CRFV" means that the adversary establishes a shadow model with Resnet-LSTM on the MSCOCO dataset to attack a target model implemented by VGG-LSTM on the FLICKR 8K dataset, which is in the **constrained** scenario.

### 4.2 Metric-based M$^4$I Attack Performance

**Implementation details.** To evaluate the image captioning performance of our target model, we apply BLEU scores (BLEU-1, BLEU-2, and BLUE-3) and ROUGE-L score as the metric, which measures the similarity between the generated text and reference text. We establish a support vector machine (SVM) as the attack model that can divide the training data and non-member data. We combine the four scores calculated by the metrics as an input vector and send it to the attack model. When these scores are further away from the maximum-margin hyperplane computed by the SVM, the more confidence our attack model can have in classifying it as a member or non-member data point.

**Unrestricted scenario.** In this scenario, the target model's dataset distribution and algorithm are available, which are the maximum amount of knowledge that the adversary can gain from the target model. The complete results are shown in Figure 3, in which we compare our attack with Random Guess. Through the attack results, we can conclude that with the maximum amount of knowledge, the attacker can infer to great extent the membership by the scores from metrics. The overall accuracy is satisfactory that the highest accuracy (81%) is achieved when the target model is trained with the IAPR dataset.

In general, under the assumption that the adversary knows the algorithm and dataset distribution of the target model, our attack is fairly strong. But there are some cases that the attack is not effective

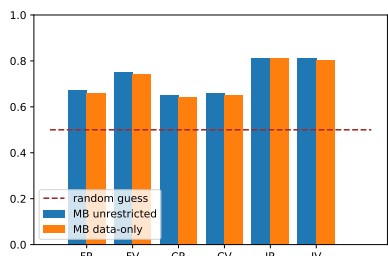

Figure 3: The metric-based attack success rate in **unrestricted** and **data-only** scenarios.

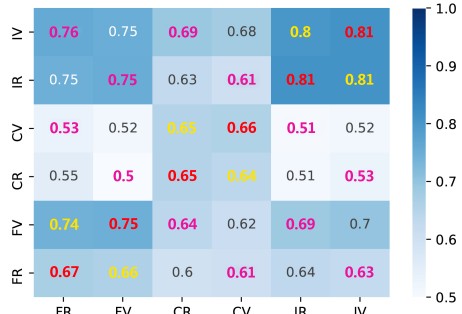

Figure 4: The metric-based attack performance in all the scenarios.

when the target model is trained with the MSCOCO dataset. There are several reasons for the ineffectiveness. The first reason is that both metrics (BLEU and ROUGE) are direct comparisons of the generated text and reference text. They cannot distinguish the difference between meaningful content and some fixed sentence patterns. For example, the image captioning model can easily generate some meaningless sentences with a fixed pattern like "This is a picture of ...". In such a case, the metrics can still give meaningless output high scores, which can cause a high false-positive rate.

**Data-only scenario.** In this assumption, the attacker can only access a shadow dataset that has the same distribution as the training dataset. The experiment results are shown in Figure 3. It is shown in the results that in this scenario, the attack performance of metric-based membership inference drops as expected but only has a minor decay compared with the **unrestricted** scenario.

**Constrained scenario.** To this end, we discuss the final and hardest attack scenario where the attacker has no knowledge about the training dataset distribution or architecture. All the experiment results are shown in Figure 4. The $x$-axis indicates the shadow model's dataset and algorithms, and similarly the $y$-axis represents the target model's dataset and algorithms. The color bar shows the attack success rate. The pink numbers indicate the attack success rate in constrained scenario. Note that all the results in **unrestricted** scenario(red numbers) and **data-only** scenario(yellow numbers) are also listed on this heat map. Under such a minimum assumption, the performance of metric-based attacks drops as expected. When the attacker does not know the model architecture and dataset information, the attack success rate of membership inference is 51% - 76%. We can find that attack performance should be highly related to the knowledge of the data distribution in the training set. As each line shows, the highest inference accuracy is always there when the attacker builds a shadow model with the data distribution same as the target model.

### 4.3   Feature-based M$^4$I Attack Performance

**Implementation details.** In our experiment, multi-modal feature extractors have been trained with the public data mentioned in Section 4.1. This feature extractor is a combination of an image encoder and a text encoder as shown in Figure 2. The attacker uses a two-layer multiplayer perceptron(MLP) as a binary classification attack model to infer the membership status. We provide more details of model architectures in Supplementary Materials.

**Unrestricted scenario.** The three assumptions are in the same condition in Section 4.2. For the unrestricted scenario, the results are shown in Figure 5. From the results, we can conclude that with knowledge of the data distribution and the architecture of the target model, the overall attack performance is better than the metric-based attack. Then, we visualize the data points (the multi-modal features of members and non-members) in two-dimensional space by t-distributed Stochastic Neighbor Embedding (t-SNE) [73]. Figure 7 shows the distributions of multi-modal features extracted by MFE from inputs and outputs of the target model and shadow model.

Without knowledge of the model architecture, attack performance drops as expected. However, in comparison with Figure 5, we can find that the accuracy decrease is even more than that in metric-based attack's performance. As can be seen from Figure 7, the multi-modal feature distributions behave slightly different when the shadow model and target model are trained with different architectures. This difference can be hardly detected by a metric-based attack, as the scores from models

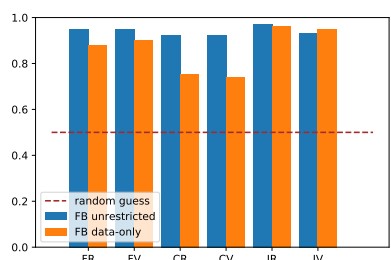

Figure 5: The feature-based attack success rate in **unrestricted** scenario and **data-only** scenario.

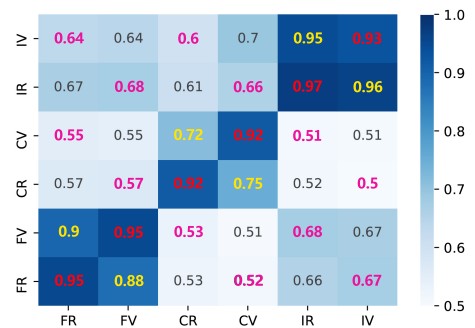

Figure 6: The feature-based attack performance in all the scenarios.

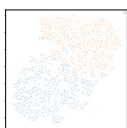 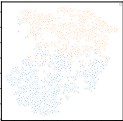 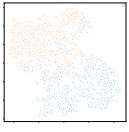 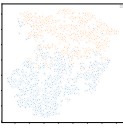 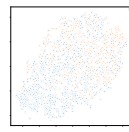 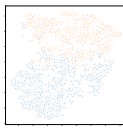

FR_shadow and FR_target       FV_shadow and FR_target       CV_shadow and FR_target

Figure 7: Visualization results by t-SNE in **unrestricted**, **data-only**, and **constrained** scenario, where blue points represent members and orange points represent non-members.

trained with different architectures are quite similar. In comparison with direct similarity calculations, the feature extractor is able to find and emphasize some inconspicuous but important differences.

**Constrained scenario.** As the hardest scenario for attackers, our attack can still achieve good performance in some cases. The experiment results are shown in Figure 6. The $x$-axis indicates the shadow model's dataset and algorithms, and similarly, the $y$-axis represents the target model's dataset and algorithms. The color bar shows the attack success rate. The attack success rate of membership inference is 50% - 68%. As shown in Figure 7, without any information, the multi-modal features of the shadow model can only have limited overlapping with the target model. Comparing with the MB attack, FB attack relies more on the knowledge of data distribution.

## 4.4 Attack Performance with False-Positive Rates

Recent works [74] and [75] mentioned that it is insufficient to use only accuracy to show the performance of membership inference attack. Here we provide the receiver operating characteristic curve (ROC curve) to show the true positive rate (TPR) versus the false positive rate (FPR) for the **unrestricted** scenario and the **data-only** scenario. From the result shown in Figure 8, it can be concluded that the feature-based attack behaves better than the metric-based attack, which means while the feature-based attack correctly picks the member data from the attacker's dataset, this method also avoids categorizing non-member data into member data.

## 4.5 Attack Performance on Medical Report Generation

To evaluate our attack, we apply our methods to the Cross-modal Memory Networks for Radiology Report Generation (R2GenCMN) [76], which is a medical report generation model by taking chest X-Ray images as input and exporting medical reports.[2] This model proposes a method with a cross-modal mapping while generating the report. We evaluate the attack performance under the **restricted** scenario, **data-only** scenario, and **constrained** scenario. Experiment setting can be seen in Supplementary Materials. The experiment results are shown in Figure 9. We find that for the medical report generation real-world task, our attack methods can still achieve strong performance. When the attacker has knowledge about both the target model's architecture and training data distribution, the feature-based attack accuracy can achieve approximately 82%.

---

[2]We directly run the model with the open-source code provided at `https://github.com/cuhksz-nlp/r2gencmn`.

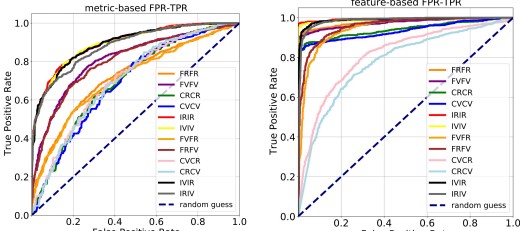
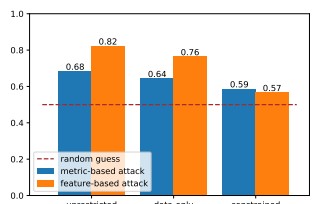

Figure 8: The true positive rate versus the false positive rate of metric-based membership inference (left) and feature-based membership inference (right). Here is the list of the **unrestricted** scenario, *i.e.*, "FRFR", and the **data-only** scenario, *i.e.*, "FVFR".

Figure 9: The metric-based attack and feature-based attack success rate on the medical report generation model. The $x$-axis indicates the attacker's background knowledge.

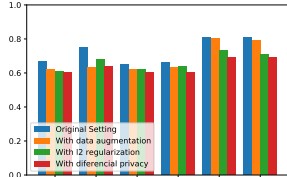
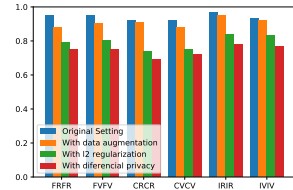
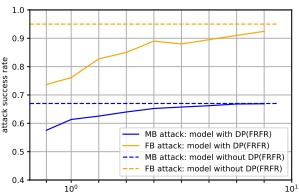

Figure 10: Comparison of attack success rate before and after applying three countermeasures. The left figure shows the results for metric-based attack and the middle figure shows the results for feature-based attack. The right figure shows the attack success rate for differential privacy training with different $\epsilon$, given $\delta = 10^{-5}$.

## 5 Mitigation

The above experiments show that our metric-based attack and feature-based attack are effective to some extent, and especially that our attack methods work well on the medical report generation tasks. Meanwhile, to defend the membership inference against multi-modal models, we also evaluate some defense mechanisms to mitigate the threat of membership leakage.

**Preventing overfitting.** When a target model overfits its training dataset, it may cause different behavior on the training and test datasets, such as higher scores in similarity metrics. An overfitting model can be weaker to the membership inference attack. The widely used techniques to reduce the impact of overfitting in machine learning are data augmentation and $l_2$ regularization [77]. From the results shown in Figure 10, we find that these two ways do help reduce the risk of membership leakage. However, the defense by data augmentation is not so efficient in some specific cases, for instance, when the target model is trained with IAPR dataset.

**Privacy enhanced training.** Differential privacy [78–82] can provide formal membership privacy guarantees for each input in the training dataset of a machine learning model. Many differentially private learning algorithms have been proposed. These algorithms add noise to the training data [83], the objective function [84, 81], or the gradient computed by (stochastic) gradient descent during the learning process [79–81]. To evaluate the attack performance with differential privacy training, we directly adopt the Opacus [85], a library for training PyTorch models with differential privacy stochastic gradient descent, on the target model. The differential privacy is set reasonably strong with $\epsilon = 1.3$, given $\delta = 10^{-5}$. In such a case, Figure 10 shows that both metric-based attack and feature-based attack performances are significantly affected. However, with differential privacy, there can be an obvious degradation in the model performance. For instance, the average ROUGE-L score for FR_target model drops from 0.245 to 0.147.

## 6 Conclusion

In this work, we take the first step to infer the data-level membership in the multi-modal models and propose two attack methods and evaluate them under different assumptions, including metric-based attack and feature-based attack. Extensive experiments show that both our attacks outperform random

guesses and the feature-based attack in general outperforms the metric-based attack. Furthermore, we adopt our attack on the pretrained medical report generation model and results show the privacy of patients is also vulnerable. Finally, we evaluate the effectiveness of data augmentation and differential private training mechanisms to defend against our attacks.

The future work can include extending our method to other domains like multi-modality such as voices or videos, extending our methods to other multi-modal tasks such as VQA, and exploring other privacy risks of the multi-modal models such as privacy leakage from the cross-modality generation models.

## Acknowledgments

This work was supported in part by facilities from CSIRO's Data61, Australia.

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
