# M$^4$I: Multi-modal Models Membership Inference

**Pingyi Hu**[*]
University of Adelaide
Australia

**Zihan Wang**[*]
University of Adelaide
Australia

**Ruoxi Sun**
CSIRO's Data61
Australia

**Hu Wang**
University of Adelaide
Australia

**Minhui Xue**
CSIRO's Data61
Australia

# Supplementary Materials

## A   Metrics Used in MB M$^4$I

**ROUGE score.** ROUGE [1] is a set of metrics comparing a produced sequence recalling a reference or a set of reference (human-produced) sequences by the ratio of reduplicated n-grams in the output and reference. ROUGE-N scores are the overlapping of n-grams [2] between the generated and reference sequence. The ROUGE-L score is one of the ROUGE-N scores with the longest common subsequence as n-grams. The range of the ROUGE score is 0-1 and a higher score means the generated sequence and ground truth sequence are more similar.

**BLEU score.** BLEU [3] is an algorithm for calculating the similarity of text. Scores are calculated for automatically generated sequences by comparing them with a set of reference sequences. BLEU scores are calculated by compute the precision of the reduplicated n-grams in the output and reference. Those scores are then averaged over the whole corpus to reach an overall quality. Intelligibility or grammatical correctness are not taken into account. BLEU-N scores are compared with n-grams [2] between the generated and reference sequence. The range of BLEU score is 0-1 and a higher score means the generated sequence and ground truth sequence are more similar.

## B   Datasets

- **MSCOCO.** The MSCOCO dataset is one of the most representative large-scale labeled image datasets available to the public. It is also the most authoritative and important benchmark in the current target recognition, detection and other fields. It contains 80 object categories around a half-million captions that describe over 330,000 images.

- **FILCKR 8k.** The FLICKR-8K dataset is a benchmark collection for sentence-based image description and search. Its image data source is Yahoo's photo album website, Flickr. Most of the images in the dataset display a human being involved in an activity. Five different captions which provide precise descriptions of the salient entities and events are supplied for each image.

- **IAPR TC-12.** The IAPR TC-12 dataset contains 20,000 still natural images globally. This includes pictures of different sports and actions, photographs of people, animals, landscapes, cities, and many other aspects of contemporary life. Each image in this dataset is associated with a text caption of English, German, and Spanish up to three different languages. We only take English captions in this study.

36th Conference on Neural Information Processing Systems (NeurIPS 2022).

## C Experiment Settings

### C.1 Data Split

**Training the target model.** We conduct our experiments on MSCOCO, FLICKR 8k, and IAPR TC-12. We randomly sample 3,000 image-text pairs from each dataset as the "member dataset" to train the target image captioning model. We use Resnet-152 as well as VGG-16, with LSTM as the architecture for the target models. For these three datasets, we randomly sample 3,000 image-text pairs from the corresponding dataset without the "member dataset" as the ground truth "non-member dataset" of the target model. Therefore, for each target image captioning model, we have 3,000 ground truth members and 3,000 ground truth non-members. For the rest of the image-text pairs left in the dataset, we randomly pick 1,000 data from each dataset as the test set to evaluate the target model and regard the remaining data as the public dataset for the attacker.

**Training shadow models.** For both proposed MMMMI attack methods, shadow models are indispensable. In the scenario where the attacker knows the target models' training data distribution, we randomly sample 6,000 image-text pairs from the corresponding dataset as the shadow dataset. In the scenario where the attacker does not have the knowledge of the data distribution, we randomly sample 6,000 image-text pairs from the MSCOCO dataset if the target model trained with FLICKR 8k dataset, randomly sample 6,000 image-text pairs from the IAPR-TC12 dataset if the target model trained with MSCOCO dataset, and randomly sample 6,000 image-text pairs from the FLICKR 8k dataset if the target model trained with IAPR TC12 dataset. With these 6,000 image-text pairs, the attacker can split them into two 3,000 image-text datasets, which can be defined as the shadow member dataset and the shadow non-member dataset. The shadow member dataset will be used to train the shadow model and the shadow non-member dataset will be used in membership inference for attack model training.

### C.2 Computational Resources

We trained our models on the NVIDIA's V100 GPU and Intel Gold 6148 CPU(40 cores @ 2.4GHz).

### C.3 MFE Model Architecture

The architecture of the image encoder is RESNET-152 with a last modified fully-connected layer, which can provide a feature with a preset shape. The architecture of the text encoder is a three-layer multilayer perceptron (MLP) which takes the one-hot encoded text as the input. We first pre-train this multimodal feature extractor on image-text pairs from the public dataset. Then, using this multimodal feature extractor to extract the multimodal feature of the shadow member data with their output from the shadow models and non-member data with the output from the shadow models as well. Then the attacker can then train a binary classification model as an attack model, which is a two-layer MLP, to infer whether a sample belongs to the target model's training dataset or not by its multimodal feature of itself and the corresponding output from the target model. The architecture of the attack model is shown as follows. The first hidden layer in the attack model has 256 units and the second hidden layer has 20 units, both activated by ReLU function. The output of the MLP is a one-dimensional output; the value indicates the probability that the sample is from a member dataset.

### C.4 Algorithm

See Algorithm 1

## D Additional Experiment 1

We have tested our MB M$^4$I and FB M$^4$I on the target model trained with the whole COCO2017 dataset. We used resnet-LSTM architecture as the target model architecture. We have trained for 50 epochs with batch-size 128. The target model behaves well on the COCO test dataset.(BLEU-1: 0.67684824, BLEU-2: 0.45550338, BLEU-3: 0.27213186, ROUGE-1:0.35076724, ROUGE-2:0.12544379, ROUGE-L:0.32091425). The final results show that the MB attack achieves 63.53% and FB attack achieves 91.15% under the **unrestricted** scenario. When the scenario is **data only**(with shadow model CV), the attack success rate can achieve 62.14% and 86.62% respectively. In the

---

**Algorithm 1** Our Feature-Based Membership Inference

---

**Require:** $f_{MFE}$ : multimodal feature extractor,
  $i_{test}$ : sample image,
  $D_p$ : image-text public dataset,
  $I_m$ : shadow member dataset,
  $I_n$ : shadow non-member dataset,
  $\mathcal{M}$ : target model,
  $\mathcal{M}'$ : shadow model,
  $\mathcal{M}_{attack}$ : attack model.
**Ensure:** Member or non-member of $I$
 1: $f_{MFE}.fit(D_p)$
 2: $T_m, T_n \leftarrow \mathcal{M}'(I_m), \mathcal{M}'(I_n)$
 3: $z_m, z_n \leftarrow f_{MFE}(I_m, T_m), f_{MFE}(I_n, T_n)$
 4: $\mathcal{M}_{attack}.fit((z_m, \text{``member''}), (z_n, \text{``non}-member''))$
 5: $t_{test} \leftarrow \mathcal{M}(i_{test})$
 6: $\mathbf{z}_i \leftarrow f_{MFE}(i_{test}, t_{test})$
 7: $y_{test} = \mathcal{M}_{attack}(\mathbf{z}_i)$
 8: **return** $y_{test}$

---

**constrained** scenario(with shadow model FV), the attack success rate can achieve 53.86%, and 52.25% respectively.

# E  Additional Experiment 2

We have evaluated our MB M[4]I and FB M[4]I on the FastSpeech2 [4], which is a SOTA text-to-speech (TTS) application that takes text as input and speech/audio as output. We directly use provided trained model from GitHub as the target model. We randomly pick 3,000 samples from its training dataset, LJSpeech [5], as members and 3,000 samples from another dataset, LibriTTS [6], as non-member samples. The shadow model shares the same structure with the target model. In the MB M[4]I method, the euclidean distance between Mel spectrograms is used as similarity metric. In the FB M[4]I settings, two 3-Layers-MLP are applied to extract features from the input text and output audio feature (from the Mel spectrograms), respectively. In this unrestricted setting, the attack success rate can achieve 86.43% and 94.24% respectively.

# F  Figures

## F.1  T-SNE Visualization

Figure 1 shows t-sne visualization results for the member and non-member distribution in all the target models and shadow models.

# G  Medical Report Generation Experiment Setting

In the **restricted** scenario, the target model is trained with the MIMIC-CXR [7] dataset. In the **unrestricted** scenario, we randomly sampled 3,000 samples from the MIMIC-CXR dataset as the ground truth member dataset and 3,000 samples from its test dataset as the non-member dataset. The shadow dataset was 6,000 samples randomly sampled from the MIMIC-CXR dataset, including 3,000 shadow members and 3,000 shadow non-members. The shadow model is trained with the same architecture on the shadow dataset. In the **data-only** scenario, the attacker trains shadow models with Resnet-LSTM architecture. While in the **constrained** scenario that the attacker does not know the data distribution and model architecture, we randomly sampled 6,000 samples from the FLICKR 8K dataset as the shadow dataset. The shadow models are trained by the Resnet-LSTM architecture on data from the FLICKR 8K dataset.

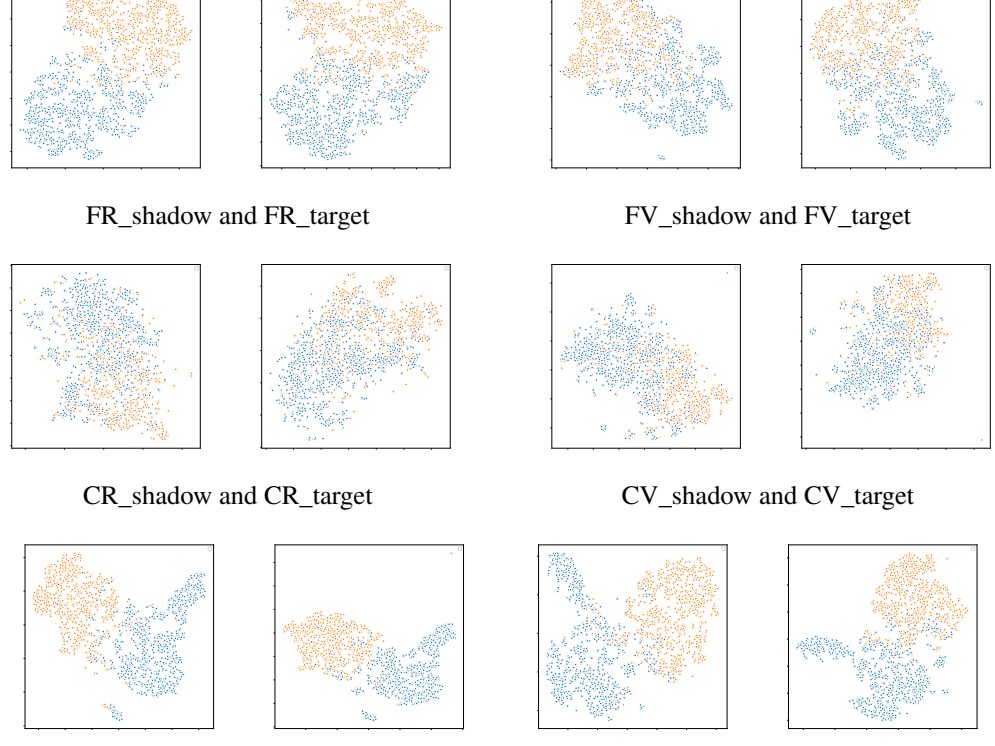

Figure 1: Visualization results by t-SNE for all the target and shadow models, where blue points represent members and orange points represent non-members.

## H  Attack Performance in Log Scale ROC

In Figure 2, we show the curve of true positive rates versus false positive rates in log scale, which is recommended by Calini et al. [8].

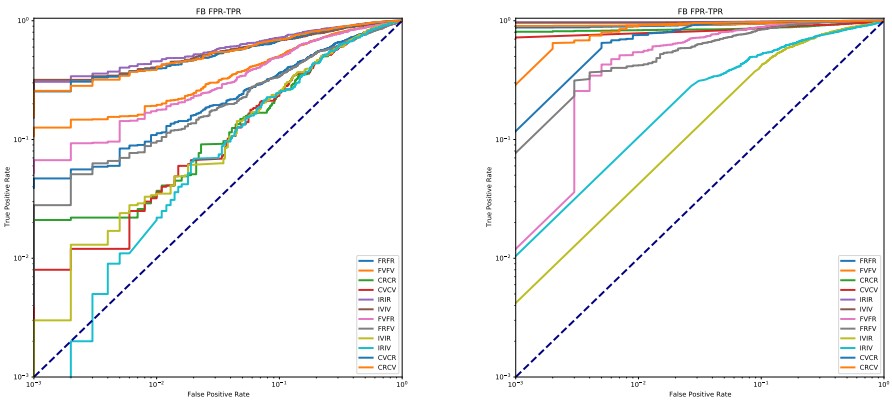

Figure 2: The true positive rate versus the false positive rate in log scale of metric-based membership inference (left) and feature-based membership inference (right). Here is the list of the **unrestricted** scenario, *i.e.*, "FRFR", and the **data-only** scenario, *i.e.*, "FVFR".

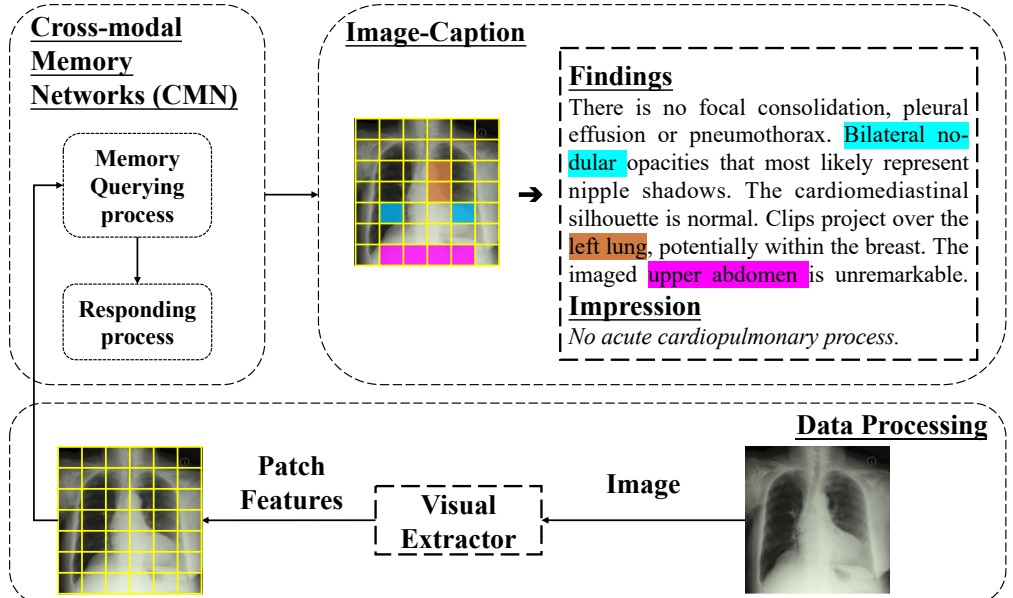

Figure 3: The overall architecture of the medical report generation model proposed in R2GenCMN.