# OpenReview forum: "M$^4$I: Multi-modal Models Membership Inference"
_NeurIPS.cc/2022/Conference — NeurIPS 2022 Accept_

### Official Review · Reviewer_BXtU · 2022-06-22

**Rating:** 9
**Confidence:** 4
**Soundness:** 4 excellent
**Presentation:** 3 good
**Contribution:** 4 excellent

**Summary:**


The paper studies the privacy leakage of multi-modal models, proposing a membership inference attack against multi-modal models. Two attack methods are introduced, the metric-based M4I and the feature-based M4I. In metric-based M4I, the adversary can score the data and use a threshold or a binary classifier to distinguish between the scores of member data and non-member data; while in feature-based M4I, a pre-trained shadow multi-modal feature extractor is used to conduct data inference attack. The experimental results show that both attack methods can achieve strong performances.

**Questions:**

Please mark the three scenarios explicitly in Figures 4 & 6.


**Ethics Review Area:**

["I don’t know"]

**Limitations:**

See my advices above.

**Strengths And Weaknesses:**


+ The topic related to multi-modal membership inference attacks is interesting and the privacy leakage of multi-modal models is an important security issue. As far as I know, this is the first research so far focusing on the membership inference of multi-modal models. Considering that the multi-modal model is one of the most rapidly rising technology in recent years, and its highly sensitive application scenarios, eg, medical, voices, or videos, the privacy risk in this domain is critical to the security community.

+ The proposed attack methods are practical and the experimental results indicate a strong performance. I appreciate the design of different scenarios (unrestricted, data-only, and constrained scenarios), which provide a thorough evaluation of the proposed attack methods. Both the metric-based and feature-based M4I are evaluated under different scenarios using different datasets. The performance is compared with the random guess, which is still reasonable,  considering that the multi-modal membership inference is a relatively new research area.

+ Mitigation of the proposed attack has been evaluated and discussed

It is always good to see the evaluation and discussion related to the mitigation of a proposed attack. Three countermeasures have been involved in the experiments. I agree that, although both metric-based and feature-based attacks are affected by the privacy-enhanced training, the trade-off can not be ignored; while the overfitting results could be a potential solution. It would be better to also discuss the trade-off of applying the defence by preventing overfitting.

I cannot find any major weaknesses that would keep me from accepting the paper but there are my two cents to further improve the paper.

- It would be nice if the three scenarios are explained and marked in Figures 4 and 6. The current figures blend the results of three scenarios in heat maps, which is good, but it could be hard for readers to compare the attack performance.

---

> ### Author Response · Authors · 2022-08-02
> **Response to Reviewer BXtU**
>
> We thank the reviewer for the valuable comments and suggestions.
>
> We will discuss the impact of applying the defense by preventing overfitting in the revised version.
>
> We have revised Figures 4 and 6 by highlighting the three scenarios, i.e. addressing different scenarios with different text colors. Explanation would be included in corresponding captions.

---

### Official Review · Reviewer_7PRw · 2022-07-11

**Rating:** 4
**Confidence:** 4
**Soundness:** 2 fair
**Presentation:** 2 fair
**Contribution:** 2 fair

**Summary:**

This paper proposes an approach for inferring the membership status in the multi-modal data, called M^4I, which aims to study the privacy leakage problem in the multi-modal learning setting. Specifically, the authors introduce two attack methods for this purpose, including a metric-based and a feature-based. Authors test the proposed methods on only a task, i.e., image captioning. The experimental results show the advantage of the proposed method.

**Questions:**

1. Will different encoders yield different results?
2. How does the structure of the feature-based model affect the results?
3. How does your model perform on other state-of-the-art image caption models?
4. Can the paper present a theory to guarantee the discrimination errors?


**Limitations:**

Yes

**Strengths And Weaknesses:**

Strengths:
1. First work to explore the membership inference problem in a multimodal setting.
2. Proposing a meaningful setting for the work.
3. Examining performance on four real-world datasets for one task.

Weaknesses:
1. The proposed models contain limited technical contributions.  For example,
1) The metric-based model only employs two existing similarity metrics on the texts as a feature for classification. Meanwhile, the feature-based model adopts an existing pre-trained model and a two-layer neural network for classification.
2) The reviewer is expected to see some theoretical results that can guarantee the discrimination errors.

2. Although the paper claims that it is under the multimodality setting, it is strange that it misses the image modality information in the metric-based model.

3. The paper only evaluates the model in one task.  It is doubtful whether it can be extended to more tasks for evaluating the generalization ability.  Therefore, it is hard to draw a conclusion that this method can perform well on other multimodal tasks.

4. The experiment section lacks a detailed description and complete comparison.  For example,
1) In Section 3.2 and the main figures, the authors mention that the feature-based model employs a pre-trained image encoder and a text encoder. However, it does not tell the exact pre-trained model in the experiment section.
2) The reviewer also doubts how the effect of the pre-trained models.  The paper lacks sufficient experimental comparisons on different pre-trained models.
3) The experiments should add more intuitive baselines to demonstrate the effectiveness of the proposed methods.

---

> ### Author Response · Authors · 2022-08-02
> **Response to Reviewer 7PRw**
>
> We thank the reviewer for the valuable comments and suggestions.
>
> * Weakness 1 and Question 4 - theoretical analysis
>
> Thanks for the good suggestion. We agree that some theoretical results will definitely strengthen the paper. However, we would like to argue that, at the current stage, our research is an empirical study, which is the first to investigate and demonstrate membership inference attacks on multi-modal models. We would like to dive into theoretical analysis in the future.
>
> * Weakness 2 - image modality information
>
> We would like to clarify that, although the metric-based attack model predicts the membership status by comparing the output and ground truth in text modality, the image modality information is still involved in the MIA, as (i) we train shadow models with the dataset of image-text pairs; and (ii) the metrics compare the output and the ground truth of the same image.
>
> * Weakness 3 and Question 3 - evaluation on SOTA
>
> We have further evaluated our metric-based attack and feature-based attack on FastSpeech2 [r3], which is a SOTA text-to-speech (TTS) application that takes text as input and speech/audio (Mel spectrogram) as output. We randomly pick 3,000 samples from its training dataset, LJSpeech [r4], as members and 3,000 samples from another dataset, LibriTTS [r5],  as non-member samples. We use all 6,000 samples to train the multimodal feature extractor in the feature-based method. The experimental results show that the metric-based attack achieves an 86.43% success rate and the feature-based attack achieves 94.24%. We will include more details about the experimental settings and results analysis in the revised version. We have considered SOTA image captioning models, such as RefineCap [r6] and RDN [r7]. As two studies [r6, r7] would be very time-consuming to implement without publicly available code and two works [r8, r9] are not reproducible due to our current computing resources, we chose to evaluate our attack on the classic encoder-decoder image captioning model [r10].
>
> * Weakness 4 and Question 2 - experiment description
>
> Due to the space limit, we provide the training details of the multimodal feature extractor in Section C in Supplementary Materials. In our experiment, the change in the structure of the multimodal feature extractor (MFE) in the feature-based method has no essential influence on our conclusion. Any MFE that can extract appropriate features should be able to work in the feature-based attack. Our research is the first step in the exploration of membership inference attacks on multimodal models.  Here we choose one usable MFE able to effectively extract the features from two different modalities for evaluation. So, we can confirm that our feature-based method is able to infer membership information. We might further study the influence on the structure of MFE. As we are the first to investigate membership inference attacks on multimodal models, to the best of our knowledge, there is no similar work that could be fairly considered as a baseline. In such a situation, we followed the approach in recent research on membership inference attacks [r11, r12] and set the baseline as random guessing.
>
> * Question 1 - encoder
>
> Different encoders in target models may yield different results. In our work, we investigate image captioning models with two different encoders, respectively based on the structure of Resnet-152 and VGG-16. The results show that the image captioning models with Resnet encoder are slightly more vulnerable to our attacks, where the attack success rate on the target model with Resnet encoder is 0.4%(in average) higher than the attack success rate on the target model with VGG encoder. The reason is perhaps, as the network structure of Resnet is deeper than VGG, the Resnet encoder may extract more representative features and thus benefits from the membership inference attack. However, the scope of our current research focuses on the empirical study of membership inference attacks on multimodal models, but it is definitely worthy of diving into this area in the future.
>
> [r3] Investigating on Incorporating Pretrained and Learnable Speaker Representations for Multi-Speaker Multi-Style Text-to-Speech. ICASSP, 2021.
>
> [r4] The LJ Speech Dataset. https://keithito.com/LJ-Speech-Dataset/, 2017
>
> [r5] LibriTTS: A Corpus Derived from LibriSpeech for Text-to-Speech. Interspeech, 2019
>
> [r6] RefineCap: Concept-Aware Refinement for Image Captioning. CoRR, 2021.
>
> [r7] Reflective Decoding Network for Image Captioning. ICCV, 2019.
>
> [r8] X-Linear Attention Networks for Image Captioning. CVPR, 2020.
>
> [r9] ClipCap: CLIP Prefix for Image Captioning. CoRR, 2020.
>
> [r10] Show and Tell: A Neural Image Caption Generator. CVPR, 2015.
>
> [r11] Membership Inference Attacks against Machine Learning Models. IEEE Symposium on Security and Privacy (Oakland), 2017.
>
> [r12] Membership Inference Attacks against Recommender Systems. ACM CCS, 2021.

---

### Official Review · Reviewer_jdYs · 2022-07-11

**Rating:** 6
**Confidence:** 3
**Soundness:** 3 good
**Presentation:** 3 good
**Contribution:** 3 good

**Summary:**

The authors proposed a Metric-based and a feature-based membership inference attack for multi-modal machine learning. The authors evaluated the  proposed attacks under unrestricted, data-only, and constrained settings on various datasets and architectures to show the effectiveness of the attacks. The authors also evaluated the effect of two mechanisms (preventing overfitting and using DP) to mitigate MIA.

**Questions:**

1. For Fig 8, shouldn’t the ROC be plotted in log-scale as Carlini etal. [73] suggested? Also, can data augmentation be used to improve the attack as in [73]?
2. For the unrestricted setting, does the shadow training dataset overlap with the target training dataset? Are there any differences in the attack success rate if the shadow training data does or does not overlap with the target training dataset?


**Limitations:**

1. In the paper, the authors mentioned the attack is not as effective when the target model is trained on MS-COCO dataset (the attack success rate is only slightly above 50%) and the reason might be that ROUGE and BLEU scores cannot capture the semantics well. Have you considered other scoring methods?
2. The authors only evaluated the proposed on image captioning tasks. It would be nice if the authors can show that the attack is successful on other multimodal applications.


**Strengths And Weaknesses:**

Strength:
The authors proposed the first MIA against multi-modal ML.
The overall presentation of the paper is well organized and easy to read.

Weakness:
The authors propose preventing overfitting (data augmentation + l2 reg) as a mechanism to mitigate MIA. These two techniques are commonly used in ML training, and I believe the attack should be evaluated on non-overfit target models.

---

> ### Author Response · Authors · 2022-08-02
> **Response to Reviewer jdYs**
>
> We thank the reviewer for the valuable feedback and address the concerns as follows.
>
> * Weakness -  evaluation on non-overfit models
>
> In Section 5 of the paper, we present the performance of applying attack on the non-overfit models (i.e. models that are enhanced by overfitting prevention methods). Moreover, in the supplementary materials, we have tested our attacks on the model that is trained on the whole COCO2017 dataset, which is a non-overfit model (performs quite well on the test set with a BLEU score achieving 0.677). The experimental result shows that the proposed attack performs well on non-overfitted models (as shown in Figure 10 and Section D in Supplementary Material).
>
> * Question 1 - ROC and data augmentation
>
> Following the suggestions from Carlini et al. [73], we report the true positive rate and false positive rate in the evaluation of membership inference attack. We will update ROC with log-scale in the revised version (as shown in Figure 2 in the updated Supplementary Materials).
> Data augmentation can be used to improve the attack. In the feature-based method, we trained the multimodal feature extractor (MFE) with data augmentation [r1]. The average attack success rate of data augmented MFE is 72.69% (in all scenarios), while the feature-based attack without data augmentation training achieves 69.51% on average (as shown in Figure 6). We will add more details of the experiment in the revised version.
>
> * Question 2 - training dataset overlapping
>
> In unrestricted scenarios, where the shadow training dataset can be overlapped with the target training dataset, the attack performance is better than that in constrained scenarios where no overlap exists, as shown in Figure 4 and Figure 6. The reason is that more overlaps between the shadow and target training datasets may lead to a better mimicking of the target model by the shadow model. Then the thresholds learned from the shadow models could be more suitable for the target model. Therefore, if more shadow training data overlaps with the target training dataset, the attack success rate can be increased.
>
> * Limitation 1 - metrics
>
> Thanks for the suggestion. We agree that it is possible for the attackers to choose more powerful metrics in our proposed metric-based attack to achieve a higher attack success rate. Considering that the BLEU scores indicate the precision of the results and the ROUGE scores indicate the recall of the results, there exists improvement space when metrics with more factors are involved. Although our research focuses on the demonstration of a metric-based MIA, it would be interesting to dive into an ablation study with more metrics considered in the future. For example, we have considered METEOR [r2], a metric based on precision and recall scores, as an additional metric, but did not use it in the final version due to its performance.
>
> * Limitation 2 - evaluate on SOTA
>
> We have further evaluated our metric-based attack and feature-based attack on FastSpeech2 [r3], which is a SOTA text-to-speech (TTS) application that takes text as input and speech/audio (Mel spectrogram) as output. We randomly pick 3,000 samples from its training dataset, LJSpeech [r4], as members and 3,000 samples from another dataset, LibriTTS [r5],  as non-member samples. We use all the 6,000 samples to train the multimodal feature extractor in the feature-based method. The experimental results show that the metric-based attack achieves an 86.43% success rate and the feature-based attack achieves 94.24%. We will include more details about the experimental settings and results analysis in the revised version.
>
> [r1] A survey on image data augmentation for deep learning[J]. Journal of big data, 2019
>
> [r2] METEOR: An automatic metric for MT evaluation with improved correlation with human judgments. ACL workshop on intrinsic and extrinsic evaluation measures for machine translation and/or summarization. 2005.
>
> [r3] Investigating on Incorporating Pretrained and Learnable Speaker Representations for Multi-Speaker Multi-Style Text-to-Speech. ICASSP, 2021.
>
> [r4] The LJ Speech Dataset. https://keithito.com/LJ-Speech-Dataset/, 2017
>
> [r5] LibriTTS: A Corpus Derived from LibriSpeech for Text-to-Speech. Interspeech, 2019

---

> > ### Comment · Reviewer_jdYs · 2022-08-08
> > **Thanks for your response. Quick follow-up:**
> >
> > For limitation 1 (about investigating better metrics), you said “it would be interesting to dive into … in the future”. But have you tried other metrics, which you can discuss (even if they did not perform very well)?

---

> > > ### Author Response · Authors · 2022-08-08
> > > **Thanks for your further question**
> > >
> > > Thanks for your comment. As we responded previously, we considered METEOR [r2], a metric based on precision and recall scores, as an additional metric, but there is not much difference in terms of performance in comparison to what the paper used. Due to little difference in terms of performance and space limitation, METEOR is excluded in this submission. We commit to discussing this metric in the final version.

---

### Meta-Review · Area_Chair_1h6m · 2022-08-29

**Recommendation:** Accept
**Confidence:** Less certain

**Metareview:**

This work studies membership inference attacks for multimodal models. It proposes a few different attacks under different assumptions on the attack model, and evaluates them empirically.
The reviewers found the problem interesting and the paper well-written. The paper is a welcome addition to the literature on membership inference attacks and should be of interest to this conference. I would encourage the reviewers to address the feedback from the reviewers in the final version.  I recommend acceptance.

**Award:**

No

---

### Decision · Program_Chairs · 2022-09-14

Accept